# Wildlife Undergrads Spread Their Wings in Citizen Science Research Experience

**DOI:** 10.3390/ijerph192416983

**Published:** 2022-12-17

**Authors:** Janel L. Ortiz, April A. Torres Conkey, Leonard A. Brennan, LaVonne Fedynich, Marybeth Green

**Affiliations:** 1Center for Excellence in Mathematics and Science Teaching and Department of Biological Sciences, California State Polytechnic University-Pomona, Pomona, CA 91768, USA; 2Caesar Kleberg Wildlife Research Institute, Texas A&M University-Kingsville, Kingsville, TX 78363, USA; 3Educational Leadership and Counseling, Texas A&M University-Kingsville, Kingsville, TX 78363, USA

**Keywords:** citizen science, science education, professional development, science communication

## Abstract

Citizen science has become a valuable tool for natural resource professionals; however, many undergraduate students are not aware of its use as a means of collecting data for scientific analysis. To address this, we introduced a bird-focused research experience into an undergraduate Wildlife Management Techniques course. The objective of this course is to provide practical experiences in wildlife science by learning and using tools and techniques applied in the field. Students designed, implemented, and presented the results of a wild bird observation survey that contributed data to an existing e-Bird-based program. Pre-post surveys were collected to assess student learning and behavioral changes. We observed increased student awareness of citizen science. Ninety-two percent of students correctly defined citizen science following the experience. However, only sixteen percent of students stated they would continue participation in STWB, fifty percent were unsure, and thirty-four percent would not continue involvement. Improvements are discussed to promote participation in citizen science, connections with the community, and communication skill development for future employment.

## 1. Introduction

Volunteer bird monitoring began in the year 1900, when ornithologist Frank Chapman suggested a new Christmas tradition of counting birds instead of hunting them, and the Christmas Bird Count (CBC) was born [1]. What started with 27 volunteers counting birds in 25 different locations across Canada and the United States [2], is now run by the National Audubon Society with tens of thousands of volunteers participating across the Americas. In the year 1966, in response to concerns about bird declines due to exposure to the pesticide dichlorodiphenyltrichloroethane (DDT), the Breeding Bird Survey (BBS) was launched by the United States Geological Survey (USGS) [3,4]. Each project collects data during a corresponding season, winter and spring, respectively, following a protocol unique to the Christmas Bird Count and Breeding Bird Survey. However, these two surveys are limited to two times per year and are missing much of the diversity and distribution of bird species. To close the seasonal gap, the Cornell Lab of Ornithology developed eBird. Created in 2003, eBird has grown exponentially in users and entries within the last decade [3,5]. The eBird platform serves as a database to submit bird sightings throughout the United States and beyond with data collected continuously throughout the year from anyone who becomes a member. The Cornell Lab of Ornithology has been a major producer of citizen science projects including eBird, BirdSleuth, Project Feederwatch, and the House Finch Disease Survey [6]. The diverse set of projects from Cornell ranges from biodiversity monitoring and disease spread among bird species, to environmental education.

Citizen science (may also be called community science; see [7]) originally included projects where volunteers collected data for project leaders with the mutual goal of advancing scientific research [7,8]. Projects involving bird monitoring often attract participants that are very familiar with the outdoors, such as bird watchers or self-proclaimed birders who consider themselves naturalists or amateur scientists, however, more projects are reaching out to those who have little to no experience in the field, including young adults and children. These projects open the doors to ornithology, offer opportunities to become engaged in science, and possibly stir an interest in STEM (Science, Technology, Engineering, & Math) fields as future careers. Citizen science as part of environmental or conservation educational programming from various agencies, non-profit organizations, and other entities offers the public the opportunity to develop their observational skills, a key component to the scientific method. It also allows the community to become aware of their surroundings, improve their knowledge of nature and appreciate natural resources through the ability of these entities to communicate science to a broader audience [9,10,11].

Young adults often have varying levels of experience and knowledge [12]. For wildlife undergraduate students this is particularly the case for ornithology skills such as bird survey methodology, bird identification, and wildlife study design. Gaps in experience in these concepts serve as an opportunity for a course project, in which undergraduate students learn about local birds, wildlife study design, and develop their scientific writing, while working collaboratively with research partners. Because wild birds are usually conspicuous in most environments, additional travel time and budgeting is not necessary for an on-campus class time experience. As students gain experience in the research process, this facilitates their potential involvement and contribution of information to a citizen science program, further expanding experiential or “learning through experience” opportunities [3,13,14,15], and providing them with valuable job skills in the process such as science communication.

Dissemination of research is a key component to advancing science. Important findings based on citizen science data and methods have made their way into many influential scientific articles including some in the most prestigious journals, e.g., [16,17]. Citizen science findings are also important to communicate from a personal and societal perspective. Individuals holding roles in professional organizations or entities are tasked with communicating with stakeholders, colleagues, and the public. In addition, part of their role may include their attendance at state, regional, national, and even international conferences to communicate their latest research findings. Locally, individuals may be asked to give a talk to a group of grade school children or civic organizations such as a Rotary Club, where those participants have some interest in the topic at hand. Being able to take scientific information, like that produced from citizen science, digest it, and present it to a variety of people from a multitude of backgrounds is an extremely valuable asset for a professional to reach a diverse audience.

The Cornell Lab of Ornithology collaborated with the Caesar Kleberg Wildlife Research Institute (CKWRI) at Texas A&M University—Kingsville (TAMUK) to develop “South Texas Wintering Birds” an online citizen science program in which participants can enter bird sightings and access data within the site. South Texas Wintering Birds (STWB) began in 2005 and was active until 2018 as an opportunity to gain more information regarding the many species of Nearctic-Neotropical migrants that winter and migrate through South Texas [18] and to encourage bird reporting by private landowners whose land is unavailable to the public. Citizen science opportunities allow private landowners, ranchers, and the public to get involved and contribute to scientific research; this has led to numerous discoveries on avian diversity, phenology, and declines, and can aid in conservation planning and general species knowledge [19,20,21,22,23].

Our main objectives were to: (1) promote involvement of undergraduate students in the South Texas Wintering Birds citizen science program and (2) improve undergraduate students’ awareness of and interest in participating in citizen science. This was implemented through a wild bird observation survey designed as a course-based undergraduate research experience (CURE). We expected students to show an increased awareness of and interest in participation in citizen science after the experience. Incorporating citizen science projects into a course experience helps link students with the broader community, to become more socially engaged with topics they may find of interest and provides an opportunity for them to develop skills for use in future employment such as communicating scientific findings.

## 2. Materials and Methods

A course-based undergraduate research experience (CURE) was implemented into an existing Wildlife Management Techniques course in Fall 2016 in the Department of Rangeland and Wildlife Sciences at Texas A&M University-Kingsville. Students designed their own bird monitoring study, conducted bird surveys, entered data into the STWB site, analyzed their data, and communicated their findings in a written report. Full details regarding the implementation of this CURE and related methods can be found in Ortiz et al. [24]. The 44 enrolled students were provided a consent form and of those, 38 students signed the consent form and completed both the written pre-survey (at the start of the course) and post-survey (at the end of the course) surrounding the implementation of the study. The pre-survey assessed their awareness of citizen science through the following yes-or-no and open-response questions:Have you heard of citizen science?○If yes, how would you define citizen science?○If yes, are you a part of any projects, and if so, which one(s)?Have you heard of South Texas Wintering Birds?

The post-survey included the above questions in addition to the following five-point Likert statement, with responses that ranged from completely disagree to completely agree:I will continue to use the South Texas Wintering Birds website.

Additional questions about gender, race/ethnicity, outdoor activity participation, and their top three wildlife careers that they hope to work in were included. Because these survey questions and statements were part of a much larger project survey from Ortiz et al. [24], reliability measures cannot be calculated for so few items and what is presented here was approached as a case study to gather views and experiences of the students [25]. Separate McNemar tests for paired, nominal data with continuity correction were used to analyze pre- and post-responses to the yes-or-no questions. Differences between genders and race/ethnicities in yes-or-no responses were not statistically analyzed due to low paired sample sizes but are still reported. Responses to defining citizen science were scored by the primary author as correct, if it included verbiage about the public’s involvement in scientific research at minimum, and incorrect if that verbiage was not included. Involvement in any other citizen science projects and outdoor activities were summarized. Wildlife career choices are presented in plotting networks to show the relationships among the responses provided by students. The Likert statement responses on the post-survey were analyzed with a fixed-effects ANOVA with gender and race/ethnicity as fixed effects and all interactions. Statistical analyses were conducted in program R. All consent forms and surveys were approved by the Institutional Review Board of Texas A&M University-Kingsville under protocol 2016-070.

## 3. Results

### 3.1. Participant Demographics, Outdoor Activities, and Career Choices

Student participant demographics including gender and race/ethnicity are shown in Table 1 with the majority of students identifying as males and White/Caucasian. A majority of students reported participating in the outdoor activities of fishing (71%) and hunting (63%). Other outdoor activities reported were birdwatching (32%), hiking (29%), kayaking or canoeing (24%), and camping (21%). Few students reported biking (11%) and photography (11%) as outdoor activities in which they participate. The top career choice reported on both surveys was game warden (pre: 24%, post: 19%), followed by wildlife biologist (pre: 14%, post: 18%), and ranch manager (pre: 14%, post: 16%) with all three often being mentioned together as the top three careers from students. Other career choices included biologist, consultant, zookeeper, wildlife manager, wildlife rehabilitator, and park ranger. Figure 1 and Figure 2 depict the career choices reported on both surveys and those that co-occurred, meaning the responses that were often reported together (ranging from one to three responses) by the same participant.

### 3.2. Awareness and Use of Citizen Science

There was a significant difference in the proportion of individuals aware (i.e., responding “yes” to hearing about citizen science) of citizen science from the pre- (*n* = 3, 8%) to the post-survey (*n* = 35, 92%) (χ2 = 33.029, df = 1, *p* < 0.001). Minimal gender differences existed. Two males were aware of citizen science at the start of the experience, whereas only one female was aware, all three reporting as White/Caucasian. Eight percent of students, the three that were aware of citizen science, attempted to define citizen science in the pre-survey, with only one student presenting a correct definition (“When citizens help scientists by gathering data”) and the other two incorrectly (“How society and its citizens view certain subjects” and “Study of people in relation to certain topics”). Ninety-two percent of students correctly defined citizen science on the post-survey, including all three students that previously attempted to define citizen science on the pre-survey. The post-survey definitions for those three students were:

“Participation and contribution to research by everyday citizens”

“When people who are not biologists record sightings or activities of animal species for broad-scale projects”

“Citizens helping gather data and helping with the scientific process”

Additional correct and incorrect example definitions written by student participants are included in Table 2. No students reported being involved in citizen science projects on the pre-survey. However, on the post-survey there were mentions of citizen science projects, including bird-window collisions (*n* = 1, 3%), which is a project that occurs in another course “Human Dimensions and Wildlife Conflict Resolution” in the department, and eBird or STWB (*n* = 18, 47%).

### 3.3. Awareness and Use of South Texas Wintering Birds

There was a significant difference in the proportion of individuals aware of South Texas Wintering Birds (i.e., responding “yes”) from the pre- (*n* = 2, 5%) to the post-survey (*n* = 33, 87%) (χ^2^ = 31.03, df = 1, *p* < 0.001). Two females were aware of STWB from the start of the experience, one reported being White/Caucasian and the other Hispanic/Latinx/Spanish. There was no significant difference in gender (F = 0.2251, df = 1, *p* > 0.05), race/ethnicity (F = 0.3323, df = 1, *p* > 0.05), or interactions when students were asked in the post-survey if they would continue use of STWB. Only 16% of students agreed they would continue using STWB, a majority of students (50%) were undecided, and 34% of students reported they disagreed with continuing their involvement.

## 4. Discussion

Students participating in the wild bird observation survey for the course had an increased awareness of citizen science and the existence of South Texas Wintering Birds, and a majority of students were able to correctly define citizen science. This was expected since students received an in-class lecture on the topic of citizen science and spent a month collecting data and entering it into the STWB platform. However, only sixteen percent of students stated they would continue participation in STWB, fifty percent were unsure, and thirty-four percent would not continue involvement. This lack of continued interest and use of this citizen science project, specifically, maybe due to a myriad of factors. The project itself was short in duration during the school semester, which may have decreased students’ investment in the project. Some students may not really care for birds in particular but prefer to focus on other wildlife species. Going beyond their own data collection for course purposes, to contribute to another format may have been seen as excessive or unnecessary in their eyes. In addition, some may have found the data entry and transfer procedures cumbersome for this website. Only two courses in the wildlife curriculum for the school year integrated citizen science-related projects, so students may not see the importance or application of it to the broader field of science and community involvement. Yet, this experience can be seen as a gentle introduction to the field of citizen science, which students are likely to revisit in their future careers.

Citizen science has been used in the classroom with students in K-12 and beyond [14,22,26,27,28]. Although fairly new, the field of citizen science has gained momentum and offers projects across a variety of fields and provides an experiential alternative to traditional lecture lessons. Volunteer bird monitoring projects such as the Christmas Bird Count, Breeding Bird Survey, and now eBird have been instrumental in getting public volunteers involved in avian monitoring, research, and providing data for scientific publications. It was not surprising that most of the wildlife undergraduate students in this study had not heard of citizen science, since it is often aimed at the public, generally non- or amateur scientists. Due to the high number of citizen science projects now available and the increased involvement of nature-based organizations in the field, it is likely that these undergraduate students may encounter citizen science upon graduation. The field is becoming a recognized, standardized, and valuable scientific research method [6]. However, in order for participation to occur, a person must be aware that these projects exist [29]. A student that is aware of and has participated in a citizen science project can list this experience as a unique and desirable skill that will stand out when applying to state agencies, non-governmental, or federal organizations that have citizen science programs in place.

### 4.1. Citizen Science as a Tool for Future Careers

A career in wildlife is often seen as an animal-first career, yet it is actually a human-first career with public communication skills as a necessity for success in the field. Citizen science is an avenue that can help students develop these communication skills to use when working with a wide-ranging audience, from scientific professionals to the general public. There has been an increasing number of jobs with duties related to citizen science within environmental agencies, non-profit organizations, and others [30]. A 18 November 2022 search for the term “citizen science” on the Conservation Job Board website (www.conservationjobboard.com; accessed on 18 November 2022), resulted in eight open positions from across the United States with job titles of outreach coordinator, educator, forest specialist, conservation director, and others. For these positions, citizen science job duties ranged from working with and training volunteers to directing citizen science programs for research and monitoring efforts. This search shows that citizen science is not restricted to education or outreach positions but has wide application across the spectrum of career choices from volunteers to directors. For example, Texas Parks and Wildlife Department (TPWD) employs two biologists that, among other duties, manage 12 iNaturalist projects for the “Texas Nature Trackers” program, to help document and monitor wild populations of plants and animals across the state [31]. Wildlife professors and graduate students at the Caesar Kleberg Wildlife Research Institute, Texas A&M University-Kingsville also engaged a regional network of 31 stakeholders across 16 counties in a citizen science project that assessed relationships between rainfall and Northern Bobwhite (*Colinus virginianus*) annual population productivity [32]. Although our students may not encounter a position solely dedicated to citizen science, they are likely to come across a citizen science project in their future if they remain on a wildlife career path.

The career of game warden was consistently listed as the top career choice among student participants, and it is a job where communication skills are critically important. This career interest is obvious throughout many of the undergraduate wildlife courses at the university. We are unsure if this is considered an ideal career because of the strong hunting culture of the region, the combination of wildlife science and law enforcement as an enticing career choice, minimal knowledge regarding wildlife jobs from high school counselors, or student’s lack of knowledge of other career options. Although we know of no use of citizen science data by game wardens in their job duties, familiarity with citizen science programs in their region could be important to officers. For example, an officer may encounter volunteers in the field and may suspect them of trespassing, poaching, or other illegal activity. Knowing about local citizen science projects and what volunteers are allowed to do, could help an officer save time in determining if illegal acts have been committed. A career as a wildlife biologist was also a top choice and has strong relations to the field of citizen science in the world of conservation careers. The National Audubon Society holds positions such as Program Coordinator for the Texas Estuarine Resource Network (TERN) Community Science Program that promotes bird monitoring and conservation through field days, restorations, and K-12 educational programs, which focus on communicating science to the public. Other organizations such as the American Bird Conservancy utilize citizen science data to learn more about bird populations like their work on the decline of three billion bird species [33]. Students also reported ranch manager as a top career choice. This may be because the state of Texas is about 98% private land with many landowners providing hunting leases for native and non-native wildlife on their properties. To our knowledge, we do not know if ranch managers utilize citizen science or its data for their land management purposes. However, projects like eBird or Monarch Watch (www.monarchwatch.org; accessed on 13 December 2022) could help a landowner identify the likelihood that a species of conservation concern is on their land, which could be helpful if they want to convert their properties from agriculture or timber state tax exemption to wildlife tax exemption.

### 4.2. Promoting Student Participation

Maintaining user participation in citizen science projects is challenging. Many factors can influence continued engagement in these projects. Marsh and Cosentino [34] found that volunteers would drop their collection routes depending on survey site characteristics being undesirable. Students in this study may have been graduating soon and knew they would no longer be able to utilize on-campus survey sites for future use or found their existing survey site to be redundant (i.e., same bird species every time) and lost interest over time. While other studies have documented strong attachment to places and increased stewardship in response to project involvement with the addition of other factors such as aesthetics of the location, personal benefits of enjoyment and relaxation, ability to contribute to science, social and learning opportunities [35,36]. Other life demands may also pull participants away from projects such as personal circumstances, constraints on time, additional commitments, associated fees, and additional effort required [36,37]. South Texas Wintering Birds was only available as a website (not a phone app) and lacked the social networking strategies that are intended to promote user interaction, communication, and retention, such as generating “likes”, followers, endorsements, earning digital recognitions or badges, and the ability to comment on other’s posts [38]. As a website that relies on self-reported observations, STWB also lacked detailed communication and feedback to users that have been shown to be very important in engaging and retaining citizen science participation [39,40].

Intrinsic and extrinsic motivation can also be a factor in the students’ continued involvement in the project. When a student continues participating in a citizen science project on their own, they are labeled as a volunteer. Volunteers are people who are unpaid, acting on free will, and acting to benefit others [41]. Clary and Snyder’s [42] theory on the functional approach to volunteering identified six motivations of volunteers including a social motivation (i.e., desire to meet new people), enhancement (i.e., wanting to improve themselves personally through the experience), understanding (i.e., wanting to learn new things), career (i.e., wanting to gain experience to benefit future careers), and others. Motivations behind a volunteer’s initial involvement have been important in how long they are likely to continue involvement compared to others [43]. Because this project was a required part of a course and not seen as students acting on their own free will, this may be a reason for disinterest in continuing involvement. However, an instructor can make stronger connections of citizen science skills to future careers to help increase the extrinsic motivation to participate or find a project that has more communication and engagement features to help with retention.

### 4.3. Beyond the Classroom

The focus of this study was the use of citizen science in the classroom, however the skills developed during this time can be applied to future careers. In the last decade, there has been an increase in the use of social media by all, particularly in America where this study took place [44]. The users of social media also include potential employers from non-profits, government agencies, and state agencies, which use these tools to engage the public and communicate not only scientific findings but educational programs, volunteer opportunities, and general information about their mission. Social media like Facebook, Instagram, YouTube, Twitter, and TikTok are all used individually or in combination to reach the public. Within these mediums is the popular use of hashtags to communicate common phrases or key terms including the increasingly popular #scicomm, short for science communication, to engage the public. Science communication as a field as grown quickly across a variety of platforms [45]. Communicating via social media requires creativity, time, and content to effectively communicate your message. Even more important in the use of social media, is the consistent engagement creating a cycle of communication between users and content creators [46]. Careers focused on social media are now broadly being advertised within wildlife organizations. A 27 January 2022 search for the term “social media” on the Conservation Job Board website (www.conservationjobboard.com; accessed on 27 January 2022), resulted in 71 open positions from across the United States with job titles of wildlife biologist, environmental technical writer, biodiversity management team coordinator, education coordinator, communication coordinator, among others. Some of the requirements of these jobs include managing the organization’s social media presence, communicating effectively with all age groups, creating content, and promoting events. These careers can create opportunities for the public to become well-informed of science activities such as how to become involved in citizen science. 

In addition, although not part of the original CURE design, we encouraged the students to communicate their research findings in the student poster session at the Texas Chapter of The Wildlife Society’s annual conference the following semester. One team of two students were interested. This shows a potential barrier and/or lack of interest in science communication with our student groups in this study. We assisted in mentoring and finding funding to support their registration and travel to present their research at the 2017 state conference. This was their first experience attending and presenting at a professional conference. The benefits of undergraduate research and conference participation on communication skills, academic performance, and career planning, particularly for students of underrepresented groups are well known [47,48,49]. 

Research, whether conducted by entire organizations, students, or by citizens, has the ability to inform policy regarding scientific issues. Findings from citizen science projects have shaped the direction in which conservation efforts have progressed. Ruiz-Gutierrez et al. [50] presented the use of eBird data by the U.S. Fish and Wildlife Service to define areas of low-risk collisions by birds in wind energy permitting. Ballard et al. [51] found that 26 museum-led citizen science programs contributed to conservation research, management, education, and policy. The programs provided data detecting river pollution, invasive species presence and eradication, under-recorded groups for species conservation assessments, along with other findings. This highlights the broader impacts that citizen science can have beyond the classroom and with the engagement of the entire community. Following the conceptual framework established by McNew-Birren and Gaul-Stout [52] and shifting the focus of this classroom project from personal engagement, concentrating on one’s own knowledge production and use based on student interests, to civic engagement can aid in expanding scientific literacy and pro-environmental behaviors of all involved.

### 4.4. Study Limitations

The citizen science aspect of this study was part of a much larger project that entailed the integration of a CURE into the classroom in Ortiz et al. [24]. Given this, the data presented here only represents a snapshot of information gathered during that project. With limited items to analyze relating to citizen science, we were unable to calculate internal consistency for the items asked of the students. This can be improved in the future by creating a separate survey specifically for evaluating the citizen science components of the project. In addition, this study was also implemented in one course section of 44 students enrolled. We were unable to have this piece integrated into other courses due to the overlapping enrollment of the same students across other wildlife courses and the instructor’s willingness to incorporate it. Furthermore, the racial/ethnic and gender identity of the students of the wildlife program is fairly uniform in each class, leaving us with the inability to find differences among the reported identities without a larger sample size. Even with these limitations, this study provided the students exposure to citizen science, with many never hearing about it previously, and opened the door for potential participation on their own, which can be replicated with university wildlife students elsewhere.

## 5. Conclusions

Challenges exist with incorporating citizen science projects into the classroom at any grade level and as shown here, a challenge exists with continuing student involvement in projects. The lack of interest in continuing involvement in South Texas Wintering Birds was somewhat unsurprising knowing that the majority of our students’ have more of an interest in large mammals and fish and STWB’s lack of engagement, feedback, and communication with users. This can also be seen in their responses to outdoor activities, with the highest-ranked being hunting (71%) and fishing (63%), while birdwatching received about half (32%) of the responses in comparison. Improvements to this class experience are dependent upon student engagement and interest, instructor motivation, and classroom flexibility. There are documented benefits and barriers to implementing citizen science into undergraduate education [53]. Here are some recommendations for those interested in integrating a citizen science project to promote ongoing student participation beyond the classroom:Choose a citizen science project with features that foster interest, motivation, incentives, feedback, and communication with users.If multiple citizen science projects are available, allow each student group to select which citizen science project they would like to work on.Incorporate data collection days throughout the quarter or semester, from beginning until the end. This will help sustain their interest and investment in the project.Hold data entry events, especially for large batches of data. As part of lecture or lab, allow students time in class to enter data into the citizen science platform. This gives students time to communicate about the project. They can ask questions and share amongst themselves what they are entering and/or seeing during their involvement with the instructor and/or peers.Incentivize participation. If their participation in the project is required, provide points that will be reflected in their grade.Have students work in groups. Providing a collaborative atmosphere gives students an opportunity for informal learning, while not having to bear the burden of all the work and further improves their communication skills. It also creates a collaboration that typically occurs in most jobs.Integrate service-learning with a local K-12 school. Partner with a local K-12 teacher to have undergraduate students teach K-12 students the basics of collecting data. This may stir interest in a broader student population and emphasize the importance of science communication with your undergraduates.Plan a citizen science day with the local community to become involved with a project alongside your students. Have your students create social media content for your organization’s outlets to practice marketing and communication skills and promote community participation before, during, and after the event.Incorporate opportunities, financial support, and mentorship for students to develop professional communication skills, such as presenting their project to lay or professional audiences.

Data collected from existing citizen science projects can also be used within the classroom and is readily available at the click of a button. As part of science education, students must learn the process of the scientific method by asking a question, designing an experiment, collecting and analyzing data, and lastly sharing the results which can be with a wide variety of audiences. With free access to data on platforms like eBird or iNaturalist, students can ask their own questions about locally relevant animals or plants within their area, go into the field and collect data, and analyze their data in conjunction with existing larger datasets online. To further engage students, an iconic animal species within your area can be chosen as a focal species and allow students to develop a passion for their local environment. An example of this can be seen within the project Operation Magpie in Australia where students learned the physical and behavioral characteristics of a common local species, the Australian Magpie [54]. These student-led projects can be used for course finals, independent research projects, or to develop their scientific skills as a way to show what students are accomplishing in the classroom. This may also encourage students to continue their involvement outside of the classroom on their own time and effort. Although students became more aware of the field and specifically the South Texas Wintering Birds project, additional changes to the implementation of such research experiences need to be met, like those mentioned previously, in order to continue to engage students in citizen science. Future research should incorporate a blended opportunity for undergraduates to work alongside K-12 students and/or the public on an existing citizen science project, to assess their interest during and beyond the class in an effort to increase their participation in society, to improve their own connection with science research, and improve the communication of their findings to the broader community.

## Figures and Tables

**Figure 1 ijerph-19-16983-f001:**
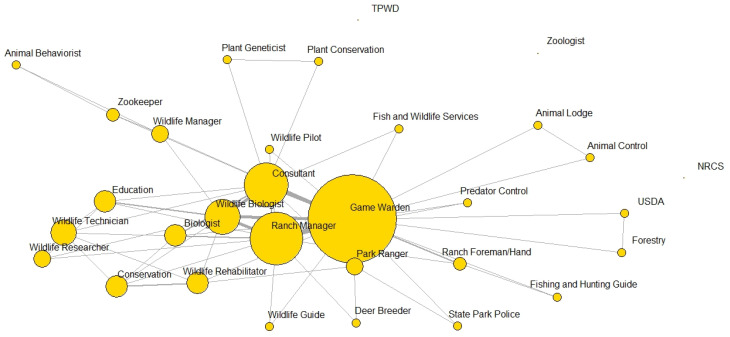
Network of career choices most reported (larger vertices) and co-occurrence of career choices (connecting lines) with bolder lines representing careers more frequently reported together on the pre-survey from students at Texas A&M University-Kingsville, Kingsville, TX, USA in Fall 2016. Note: Unconnected careers represent those mentioned on the post-survey yet were not reported by any student on the pre-survey. NRCS, Natural Resources Conservation Service; TPWD, Texas Parks and Wildlife Department; USDA, United States Department of Agriculture.

**Figure 2 ijerph-19-16983-f002:**
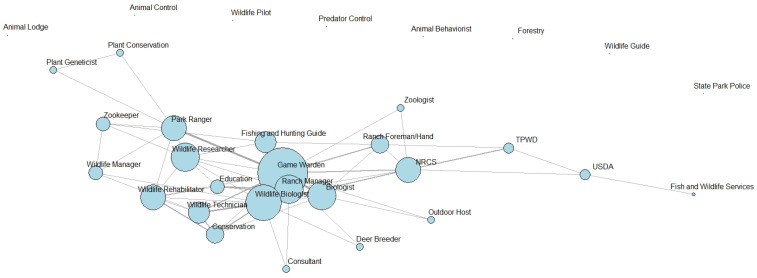
Network of career choices most reported (larger vertices) and co-occurrence of career choices (connecting lines) with bolder lines representing careers more frequently reported together on the post-survey from students at Texas A&M University-Kingsville, Kingsville, TX, USA in Fall 2016. Note: Unconnected careers represent those mentioned on the pre-survey yet were not reported by any student on the post-survey. NRCS, Natural Resources Conservation Service; TPWD, Texas Parks and Wildlife Department; USDA, United States Department of Agriculture.

**Table 1 ijerph-19-16983-t001:** Reported gender and race/ethnicity of 38 total student participants in the citizen science experience at Texas A&M University-Kingsville, Kingsville, TX, USA in Fall 2016.

Student Demographics		*n* (%)
Gender	Female	12 (32%)
	Male	26 (68%)
	Self-Identified	0 (0%)
Race/Ethnicity	Asian	0 (0%)
	Black or African American	0 (0%)
	Hispanic or Latinx or Spanish	11 (29%)
	Native American or Alaska Native	0 (0%)
	Native Hawaiian or Other Pacific Islander	0 (0%)
	White or Caucasian	27 (71%)
	Other	0 (0%)

**Table 2 ijerph-19-16983-t002:** Example student definitions of citizen science provided on pre- and post-surveys at Texas A&M University-Kingsville, Kingsville, TX, USA in Fall 2016.

Score	Example Definition
Incorrect	A way for citizens who love birdwatching, may use their skills to identify birds in a given area, and record bird sightings.
	Wildlife techniques class at TAMUK.
	Act of observing how people interact with their surroundings.
Correct	Citizens helping with research projects.
	Everyday people doing research/data work to assist in a larger project.
	Citizens doing their part to develop research that will further help scientific progress.

## Data Availability

The data presented in this study are openly available in Mendeley Data at DOI: 10.17632/7jkmnbb9gc.1.

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
