# Peer review of "Wildlife Undergrads Spread Their Wings in Citizen Science Research Experience"

_ijerph, 2022, doi:10.3390/ijerph192416983_

Round 1
Reviewer 1 Report
The conclusion section of the manuscript, "recommendations for those interested in integrating a citizen science project", is what will be most useful to others. I would expand on that section, if possible, as it is a lot more interesting than details about survey demographics.
In case this suggestion would be helpful, I would add this bullet point to that list of suggestions: "If multiple citizen science projects are available, allow each student group to select which citizen science project they want to work on." This might help with some of those 34% who wouldn't continue involvement.
Line 93 mentions that the "(STWB) began in 2005 and was active until 2018". I was curious why it ended.
Figures 1 & 2 are interesting, but very low resolution in my PDF. Hopefully the final version looks better, as it is very pixelated if you zoom in to try to read anything.
I did not find a reference for Monarch Watch on line 297, and it would be nice to have a citation for "more recently called community science" on line 51.
Author Response
Thank you for taking the time to review our manuscript. We believe your recommendations have greatly improved it. Please see attachment.

Reviewer 2 Report
This is an interesting and well written manuscript that highlights how citizen science can be implemented into tertiary education. This study is therefore relevant to both citizen science research, but also to higher education research. I have a few minor suggestions to improve detail in sections.
Line 53-63- While citizen science is introduced here, it would be useful to also include one very specific definition of citizen science.
Line 78-80: Need to reword this sentence as starting with “However, citizen science findings are still important” does not flow appropriately from the previous sentence
Line 119- It would be useful to state when the pre and post surveys were collected (i.e. at the start of the course?)
Line 130- It is not clear what data were collected via the question “top three wildlife careers”- is this the careers that they hope to work in?
Lined 180 & 205- Should not use terminology “highly significant” and instead use “significant”
Line 337: The line that discusses social media use increasing needs to be referenced
Line 363- Could expand on the finding that only 1 group choose to communicate their research as it shows there is a barrier to science communication for most of the groups in this study
Author Response

(The authors gave the same response as above.)
